# Smoking Cessation Rates among Patients with Rheumatoid Arthritis and Osteoarthritis Following the ‘Gold Standard Programme’ (GSP): A Prospective Analysis from the Danish Smoking Cessation Database

**DOI:** 10.3390/ijerph19105815

**Published:** 2022-05-10

**Authors:** Monika Laugesen, Mette Rasmussen, Robin Christensen, Hanne Tønnesen, Henning Bliddal

**Affiliations:** 1Clinical Health Promotion Centre, WHO-CC, The Parker Institute, Bispebjerg and Frederiksberg Hospital, University of Copenhagen, 2000 Copenhagen, Denmark; monika.laugesen@hotmail.com (M.L.); mette.rasmussen.03@regionh.dk (M.R.); hanne.tonnesen@regionh.dk (H.T.); 2Clinical Health Promotion Centre, WHO-CC, Department of Health Sciences, Lund University, 20502 Malmö, Sweden; 3Section for Biostatistics and Evidence-Based Research, The Parker Institute, Bispebjerg and Frederiksberg Hospital, 2000 Copenhagen, Denmark; robin.christensen@regionh.dk; 4Research Unit of Rheumatology, Department of Clinical Research, University of Southern Denmark and Odense University Hospital, 5000 Odense, Denmark

**Keywords:** rheumatoid arthritis, osteoarthritis, smoking, intensive smoking cessation intervention, Gold Standard Programme, national database

## Abstract

(1) Background: Smoking cessation may be very difficult, even if smoking aggravates the prognosis of a disease, which has been shown to be the case for persons with rheumatoid arthritis (RA). In contrast, an association in patients with osteoarthritis (OA) is still disputed. The primary objective was to compare smokers diagnosed with RA and OA to controls, regarding smoking cessation rates after following the intensive ‘Gold Standard programme’ (GSP). Secondary objectives included the identification of significant prognostic factors for successful quitting. (2) Methods: In total, 24,652 patients were included in this prospective cohort study, after attending the national GSP for smoking cessation intervention 2006–2016, as registered in the Danish Smoking Cessation Database. Data were linked to the National Patient Register. Hereof, 227 patients (1%) were diagnosed with seropositive RA and 2899 (12%) with OA. Primary outcome was continuous abstinence six months after the planned quitting date. (3) Results: In total, 16,969 (69%) of the patients participated in the follow-up interviews. The adjusted odds ratios for successful quitting were similar to the control group for both RA (1.28, 95% CI: 0.90–1.80) and OA patients (0.92, 0.82–1.03). The outermost, strongest positive factor for successful quitting was compliance, defined as attending ≥75% of the meetings. To a lesser degree, attending an individual intervention was a positive predictor, while being heavy smokers, disadvantaged smokers, women, living with a smoker, and if GSP was recommended by health professionals were negative predictors. (4) Conclusions: The odds ratios for quitting were similar to controls for both RA and OR patients. Additional research is needed to determine effective actions towards increased attendance at the programmes.

## 1. Introduction

Smoking tobacco is defined by the WHO as one of the biggest health issues in the world, killing more than seven million people every year [1], and it is on top of the list of preventable risk factors worldwide [2]. In Denmark, 14% of the population was registered as a daily smoker in 2021 [3].

Smoking cessation may be difficult and require intensive intervention programmes, even if smoking aggravates the prognosis of a disease, which has been shown to be the case for persons with rheumatoid arthritis (RA). Though smoking has not been proven causal to the development, it has been described to exacerbate RA [4], characterised by higher disease activity, more pain, lower health-related quality of life [5], higher rheumatoid factor titers, use of disease-modifying anti-rheumatic drugs [6], more morning stiffness, and swollen joints [7]. In contrast, smoking cessation is related to lower disease activity [8]; however, the efficacy of smoking cessation programmes among smokers with RA seems inconclusive in a recent systematic review [9].

The association is less clear for osteoarthritis (OA)—with conflicting results reported in multiple meta-analyses: an association has been rejected in several meta-analyses [9,10,11,12], whilst another contradicts this, indicating an inverse association between smoking and the risk of developing knee-OA [13]. Furthermore, a Swedish study found an overrepresentation of smokers among patients with OA [14]. Therefore, smoking and smoking cessation are still under consideration as prognostic factors, with a negative impact to some degree for OA.

Based on the association between smoking and RA, as well as the possible association with OA, it would be relevant to investigate the successful quit rate for those patients, which is possible via the unique national Danish Smoking Cessation Database (SCDB). Since 2001, a total of 423 clinics have registered data for more than 150,000 smokers attending the standardised intensive smoking cessation interventions (ISCI), which are offered with no need for referral and are free of charge [15,16].

The aim of this study was to compare smokers diagnosed with RA and OA to controls regarding successful quitting after following the intensive ‘*Gold Standard Programme*’ (GSP), and, secondly, to identify significant prognostic factors.

Due to frequent smoking among patients with RA and OA, more difficulty in quitting may be expected, and our main hypothesis was, therefore, that persons with RA or OA have lower quit rates than persons without those diagnoses.

## 2. Methods

### 2.1. Study Design

This prospective cohort study on the effect of a specific ISCI was based on timely collected data from the Danish SCDB, a research database designed and registered to evaluate the effectiveness of smoking cessation interventions for different groups of smokers [15]. To compare quit rates of patients with arthritis to that of the control group, consisting of non-arthritis patients, data from the SCDB were linked to the National Patient Register (NPR) [17,18] using a personal and unique ten-digit number (CPR) [19]. All hospital-related data are recorded in the NPR at any hospital contact, including diagnoses, based on the International Classification of Diseases, 10th edition (ICD-10) [18]. This study was reported according to the RECORD guidelines (REporting of studies Conducted using Observational Routinely-collected Data) [20].

### 2.2. Setting

All Danish citizens have access to SCI clinics, which have no referral required and are free of charge [15]. The clinics are localised at municipal clinics, pharmacies, hospitals (including midwiferies), general practitioners, and other private companies, and it is estimated that 80–90% of all individuals participating in an SCI are reported to the Smoking Cessation Database [15]. All participants gave informed consent for registration and follow up. The study was approved by the Danish Data Protection Agency (2010-41-5463/2000-54-0013) and the Scientific Ethical Committee (H-C-FSP-2010-049).

### 2.3. Participants

Adult patients were included in this study if they were registered in the SCDB, having participated in ISCI between January 2006 and December 2016, with follow up in 2017. All patients were daily smokers at the beginning of the ISCI. If a person attended more than one ISCI, only the latest was included. All patients were cross-referenced with the NPR to identify the presence of arthritis (DM00-DM19) [21]. Arthritis diagnosed before initiation of the ISCI was included. Patients were dichotomised with or without ICD-10 codes DM00-99. Patients without any of the above-mentioned diagnoses represented the control group. Patients diagnosed with DM00-99 were categorised into DM05 (seropositive RA), DM15-19 (OA), and other DM-diagnoses. Patients with other DM-diagnoses were excluded. DM06 (seronegative RA) was estimated to be less well defined in comparison to DM05, so it was not included. If a patient was diagnosed with more than one arthritis diagnosis among DM05 and DM15-19, DM05 was superior to DM15-19 and chosen as the main diagnosis.

### 2.4. Intervention

The Danish ISCI is considered the Gold Standard Programme (GSP) and delivered (manual-based) by trained counsellors. GSP consists of 5–6 group- or individual-based face-to-face meetings over 6 weeks, combined with tailored nicotine replacement therapy individually chosen by the participants or similar supportive medicine, according to the Fagerström test score on nicotine dependence [15]. The total intervention time was approximately 2 or 10 h for individual or group interventions, respectively. This includes motivational dialogues, a patient education programme, and a follow up for successful quitting at the end of the programme and again 6 months after the quit date [15]. Additionally, a hotline was available Monday-Friday during work hours [22].

### 2.5. Outcomes

The main outcome was successful quitting, measured as self-reported continuous abstinence throughout the 6 months of follow up.

### 2.6. Data

Baseline sociodemographic data: Age categorised as 18–49, 50–64, or 65 or older; sex; being a disadvantaged smoker; defined as unemployed; and/or having short or no education (no more than 12 years of school and short work-related courses).

Baseline smoking profile: Heavy smoking: ≥20 pack-years, ≥20 cigarettes daily, and/or Fagerström test for nicotine dependence ≥7 points (ranging 0–10, with zero being no dependency), previous recurrence of smoking after cessation attempts within the last 10 years (smoke-free for at least 14 days), and living with a smoker.

Variables concerning the ISCI: Setting, format, free nicotine replacement therapy, or other supportive medicine, recommendation to quit by a health professional and compliance to the GSP in terms of meeting attendance. Full compliance was defined as attending at least 75% of the planned meetings [23].

Follow-up data: Patients were contacted by phone to follow up on their smoking status 6 months (±1 month) after their planned quit day. Smoking status was reported dichotomously and collected by qualified personnel through structured phone interviews. If participants were not reached after at least four phone call attempts, at least one being in the evening, they were considered non-respondents [15]. Patients who did not want to be contacted; were reported dead, missing, or emigrated at follow up; had a missing smoking status; or were not reached at the time of follow up (non-respondents) were considered lost to follow up.

Age was collected as a continuous variable. Cigarettes per day, years of smoking, and Fagerström score were collected as discrete data, while all other covariates were collected as categorical variables.

### 2.7. Data Access and Cleaning

The researchers had full access to data recorded in the Smoking Cessation Database in the study period. As the CPR number was the unique key used to identify participants and link data on a person level from the LPR, all CPR numbers were checked using validation rules. The Civil Registration System was used to correct an invalid CPR. If correction was not possible, the smokers were excluded from the database.

Dates were automatically validated upon entry in the SCDB, while daily consumption of tobacco and years of smoking were manually checked. Data were recoded to missing, if a patient was using >100 g of tobacco per day, as this was not considered likely, or if a patient reported years of smoking as greater than their age. From the NPR, the researchers had access to all hospital contacts (in- or outpatient) registered from 1995 through August 2017, for any of the validated CPR numbers.

### 2.8. Statistical Methods

Since the outcome variable was dichotomous, we used logistic regression analysis models. Data were analysed using both univariable and multivariable models, presenting results as odds ratios (ORs) and the corresponding 95% confidence intervals (95% CI). The exposure variable was being diagnosed with that of the arthritis, and the predictor variables included age, sex, being disadvantaged, heavy smoking, previous quit attempts, living with a smoker, intervention setting, format, compliance to the GSP, free NRT or other supportive medicine, recommendation by health staff, and calendar year of intervention. All analyses were two sided; *p*-value < 0.05 was considered statistically significant. We did not apply explicit adjustments for multiplicity, rather, we analysed and presented all secondary analyses in prioritised order (i.e., presented in the order of prespecified importance).

Initially, univariable analyses were performed on the presence of arthritis and on each of the predictor variables listed above. The final multivariable mixed-effect regression model was, then, fitted, taking the univariate analyses and established knowledge into account and adjusting for hierarchical clustering of the smoking cessation clinic. The following variables were included in the final model: being diagnosed with arthritis, age, sex, being disadvantaged, heavy smoking, living with a smoker, format, compliance to the GSP, and recommendation by health staff. The main analyses were based on respondents (as observed) only.

A complete case analysis was used, excluding patients from the analysis if at least one variable was registered as missing. Here, the data are assumed missing completely at random (MCAR). Missing data on smoking status were assumed to be Missing Not At Random (MNAR). Sensitivity analyses were performed using a worst/best-case approach, comparing respondents (as observed) to results where those lost to follow up were all assumed to be unsuccessful quitters (worst case, M_y_m_ = 0) or were all assumed to be successful quitters (best case, M_y_m_ = 1) [24].

The proportion of successful quitters were calculated for respondents (as observed), as well as for all the worst/best-case scenarios, and Rubin’s rule, designed to pool these parameter estimates, was applied to estimate the crude quit rates. Stata/IC v.16 (StataCorp) was used for the statistical analyses.

## 3. Results

As illustrated in the flowchart in Figure 1, the number of unique smokers from the SCDB fulfilling the inclusion criteria was 38,776. These smokers were cross-referenced to the NPR using the unique CPR number to identify arthritis. After excluding 14,124 smokers who with diagnosed with “other arthritis”, we ended up with a study cohort of 24,652 smokers.

A diagnosis was identified of RA in 227 (1%) and OA in 2899 (12%) patients of the total study cohort (Figure 1). The characteristics of the groups are presented in Table 1. The groups were considered different, if the dissimilarity was greater than five percentage points compared to the control group. This was the case for both RA and OA regarding age, disadvantaged smokers, heavy smoking, being recommended by health staff, getting free NRT or other supportive medicine, and programme format. In addition, RA differed from the control group regarding sex and living with a smoker.

Patients with RA were more often women, and they were less likely to live with a smoker. Patients with RA and OA were of older age (≥50) than the control group, and a difference in both patient groups was also seen regarding more heavy smokers, disadvantaged smokers, individual programmes, and recommendations by a health professional.

The numerically largest differences were seen in age, being a disadvantaged smoker, and heavy smoking, in descending order. More specifically, the proportion of patients with RA and OA aged 65 or more was 20 percentage points and 21 percentage points higher, respectively, compared to the control group.

### 3.1. Successful Quitting

The highest crude quit rate, measured as continuous abstinence for six months, though not statistically significant, was observed in patients with RA (Figure 2). Rubin’s rule was applied to pool the imputed datasets into one estimate of the crude quit rate for each patient group. This resulted in quit rates of 39–43% (Figure 2).

### 3.2. Prognostic Factors of Importance for Successful Quitting

There was no difference in either the crude or the adjusted odds ratios (OR) of patients with OA or RA compared to the control group (Table 2). Based on the adjusted analysis, the predictors that enhanced the odds of successful quitting were being 50–64 years of age and attending an individual SCI, while the most sizeable difference was being compliant to the programme (OR: 3.24 [95% CI; 2.99–3.51]). The predictors that impaired successful quitting were being a woman, a disadvantaged or heavy smoker, living with a smoker, or being recommended to stop smoking by a health professional.

### 3.3. Sensitivity Analysis

Of the study cohort 7683 (31%) were lost to follow up, and those not responding to follow-up phone calls constituted of the largest part (6647). Statistically significant differences between respondents and non-respondents were found for the following variables: being diagnosed with arthritis, age, sex, being disadvantaged, heavy smoking, format, being compliant to the GSP, and being recommended by health staff. Those differing more than 5 percentage points were being disadvantaged (6–7 percentage points) and non-compliant (14 percentage points)—both factors were highest in being lost to follow up.

Based on an intention to survey approach, all participants were included in the sensitivity analyses. The crude and adjusted regression analyses were repeated for the worst/best-case scenario (see Appendix A, Table A1 and Table A2). In general, the model was robust, and only minor differences were observed. In the best-case scenario, smokers diagnosed with OA changed into being significantly less likely to become successful quitters (OR: 0.91 [95% CI: 084–1.00] *p* = 0.045) compared with the control group. Only regarding age, and only in the best-case analysis did, the results change direction.

## 4. Discussion

Interestingly, we could not confirm our hypothesis, as we found no statistically significant difference regarding successful quitting after six months among patients with OA or RA than among the controls, despite a higher frequency of heavy and disadvantaged smokers in the arthritis groups. The outmost important positive predictor was being compliant. The other significant predictors were only impactful to a lesser degree; the positive predictors were older age or attending an individual programme, while the negative predictors were being a heavy or disadvantaged smoker, a woman, recommended to attend by health professionals, or living with a smoker.

The benefits of smoking cessation for patients with RA are obvious, and public awareness of this fact has been substantiated by campaigns from, e.g., the Danish Rheumatism Association. Besides exacerbating RA symptoms, smoking, also, has a considerable impact on mortality beyond the risk for non-RA smokers, a risk which may be countered by smoking cessation [25]. In our study, as many as 75% of the RA smokers had smoked 20 pack-year or more.

The association between smoking and OA is disputed, however, for common health issues, so it would be recommendable for patients with OA to attend an ISCI such as the GSP to quit smoking. Our results showed that patients with OA or RA were no less successful in quitting than the control group. Varying results have been reported, with SCI tailored for patients with arthritis. An arthritis-specific programme in Spain only found little success—a quit rate of 14%—across groups with different arthritis diagnoses, though without a control group, and with a less-intensive intervention than the GSP [26]. In contrast, a small Swedish study on an RA-specific programme obtained a quit rate of 43%, when offering smoking cessation counselling conducted by a rheumatology nurse [27]. Finally, a study from New Zealand showed no difference between the RA-specific and the non-specific programmes [28]. No OA-specific programme has been published. Similar to other studies on ISCI, we saw an indication of a negative influence of being encouraged to join the GSP by health professionals [29]. The reason for this may be that health professionals mainly suggest this programme for the most heavy and disadvantaged smokers. Compliance with the GSP is highly beneficial, and each visit almost doubles the quit rates [23]. In addition, participants highly motivated to quit have a better success rate, although motivational evaluation was not part of this study. In the future, it is suggested to evaluate the impact of different strategies for further recommendations, including different groups of health professionals, non-health professionals, and combinations thereof, especially aiming at disadvantaged and vulnerable smokers.

Due to the heavy negative influence of smoking on health, all clinicians should recommend and refer smokers to highly effective smoking cessation interventions, such as the ISCI. Especially, patients with RA should be informed of the extra benefit of relieving symptoms by quitting, as these have been described to be barriers for smoking cessation [30]. This may require updated knowledge and training of health staff for implementation in everyday practice.

Dealing with heavy and disadvantaged smokers, it appears that additional care is needed, as quit rates are lowest in these groups. The Danish Ministry of Health has already brought this inequity into focus, and, in 2014, introduced free medical support for the vulnerable groups. The good effect was seen in the follow-up project: “Enhanced effort among heavy smokers 2014–2018”, where 72% of those included completed the smoking cessation programme, and 47% of these quit smoking [31].

### Strengths and Limitations

Loss to follow up and missing data, for various reasons, are difficult to avoid in large prospective register studies like this. Still, a relatively high rate of follow up in the SCDB of the national intervention programme [15] makes it possible to perform studies in the NPR, and both registers are known for their high-quality data collection [17,18]. Our findings are based on a relatively small number of patients with RA and OA, so should be substantiated in studies containing more patients.

The primary outcome was successful quitting, reported as continuous abstinence instead of point prevalence, which could be considered a strength [32]. The quit rate was obtained by interviews and not validated by, e.g., a cotinine or carbon monoxide test [33]. It would have been preferable to use a systematic CO measurement for validation, considering the individual differences [34]. Self-reported outcomes have been reported to overestimate the quit rate by 3–6% [35,36], though this limitation can be questioned, as one study found no difference in self-reported abstinence and measured cotinine level in urine [37]. Provided there was a difference in outcomes, we could expect this to have appeared in all groups, equally. The data on recommendation by health professionals to quit smoking were also self-reported and were without information on time from recommendation to entering the GSP. Furthermore, it is not known if the group not recommended by health professionals has been recommended by families, friends, workplace, other smokers, and patients, or if they were participating on their own initiative. Those lost to follow up posed a potential bias, exaggerating the quit rates, as was also seen in our analyses.

Only patients in contact with a hospital, registering arthritis as a primary or secondary diagnosis, are included in NPR [17,18], and, by consequence, this may pose a limitation regarding the generalization of the results to other settings. We inferred that the diagnosis, RA, was more valid in the case of a seropositive disease, so we did not include seronegative RA (DM06) in the analysis [38]. Similarly, the diagnosis of OA may only be used at the hospital in relation to more severe cases. Thus, for both RA and OA, it may be suspected that our patients have a more severe disease than patients followed by general or specialist practitioners.

## 5. Conclusions

Patients with OA or RA had comparable odds to the control group for successful quitting, after adjustment for possible confounding variables. The treatment adherence was of outmost importance and should be kept in focus. Special support should be provided to disadvantaged or heavy smokers, and attention should be paid to increasing compliance. More than one in three patients in the study population managed to quit for the longer term. According to our findings, it is, therefore, recommendable for patients with OA or AR to attend an ISCI as the GSP. Generalization should, however, be considered carefully regarding local conditions.

From a societal perspective, it is important to support the implementation of ISCI. By increasing the number of patients with arthritis attending a GSP, morbidity and mortality could be lowered, resulting in a substantial economic benefit for the society at large. Additional research is needed to determine which actions are effective towards increased attendance at the programmes, and how to obtain even higher quit rates of patients with RA and OA.

## Figures and Tables

**Figure 1 ijerph-19-05815-f001:**
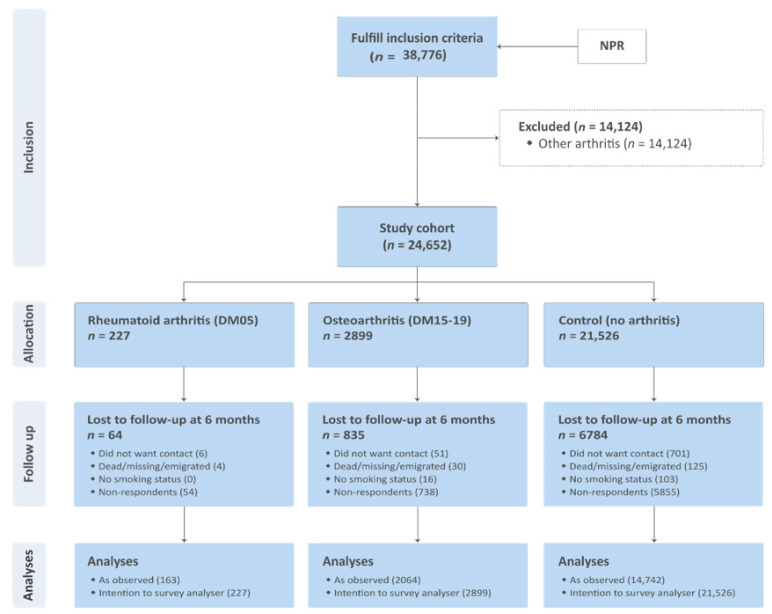
Flowchart for inclusion, allocation, follow up, and analysis. Allocation groups are based on ICD-10 codes, International Classification of Diseases, 10th edition [21]; RA, seropositive rheumatoid arthritis; OA, osteoarthritis. NPR: National Patient Register.

**Figure 2 ijerph-19-05815-f002:**
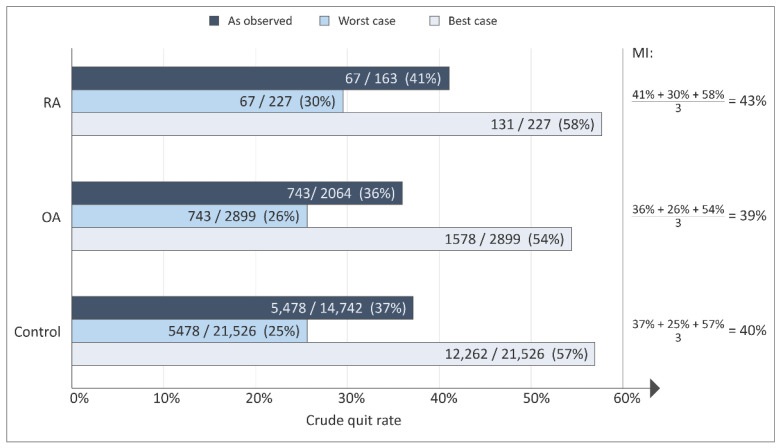
Crude quit rates of subgroups according to ICD-10, International Classification of Diseases, 10th edition [21]; RA, seropositive rheumatoid arthritis DM05; OA, osteoarthritis DM15-19; Control, non-arthritis patients; and outcome measures for successful quitting. MI: multiple imputation.

**Table 1 ijerph-19-05815-t001:** Characteristics of the participants in the study cohort according to arthritis diagnosis: RA, OA, and control (no arthritis) at baseline.

Characteristics	RA		OA		Control	
	N = 227	(1%)	N = 2899	(12%)	N = 21,526	(87%)
Participants	N	%	N	%	N	%
**Age (years)**						
18–49	46	(20)	475	(16)	11,815	(55)
50–64	109	(48)	1459	(50)	7098	(33)
65+	72	(32)	965	(33)	2613	(12)
**Sex**						
Men	58	(26)	1207	(42)	9015	(42)
Women	169	(74)	1692	(58)	12,511	(58)
**Disadvantaged smoker ^a^**						
No	99	(44)	1441	(50)	12,986	(60)
Yes	121	(53)	1323	(46)	7729	(36)
Unknown	7	(3)	135	(5)	811	(4)
**Heavy smoker ^b^**						
No	41	(18)	355	(12)	5451	(25)
Yes	178	(78)	2471	(85)	15,628	(73)
Unknown	8	(4)	73	(3)	483	(2)
**Compliance with programme ^c^**						
No	86	(38)	1029	(35)	8249	(38)
Yes	139	(61)	1845	(64)	12,976	(60)
Unknown	2	(1)	25	(1)	301	(1)
**Living with a smoker**						
No	163	(72)	2040	(70)	14,191	(66)
Yes	62	(27)	832	(29)	7150	(33)
Unknown	2	(1)	27	(1)	185	(1)
**Previous quit attempts**						
No	90	(40)	1083	(37)	8495	(39)
Yes	130	(57)	1733	(60)	12,576	(58)
Unknown	7	(3)	83	(3)	455	(2)
**Recommendation by healthcare staff ^d^**						
No	60	(26)	746	(26)	8716	(40)
Yes	153	(67)	2030	(70)	11,770	(55)
Unknown	14	(6)	123	(4)	1040	(5)
**Free NRT or other supportive medicine**						
No	113	(50)	1328	(46)	8957	(42)
Yes	102	(45)	1362	(47)	10,034	(47)
Unknown	12	(5)	209	(7)	2535	(12)
**Smoking cessation clinic**						
**Setting**						
Municipality	175	(77)	2342	(81)	16,693	(78)
Hospital (incl. midwives)	15	(7)	180	(6)	1495	(7)
Pharmacy	34	(15)	342	(12)	2821	(13)
Other	3	(1)	35	(1)	517	(2)
**Smoking cessation intervention**						
**Programme format**						
Group	176	(78)	2205	(76)	18,227	(85)
Individual	51	(22)	694	(24)	3299	(15)

RA, seropositive rheumatoid arthritis; OA, osteoarthritis; Control, non-arthritis patient (based on ICD-10, International Classification of Diseases, 10th edition [23]; DM-diagnoses: RA DM05, OA DM15-19, Control no DM00-99). ^a^ Disadvantaged: ≤12 years of school and/or unemployed. ^b^ Heavy smoker: ≥20 pack years, Fagerström score of ≥7 points, and/or daily consumption of ≥20 cigarettes. ^c^ Compliance: attended ≥ 75% of the planned meeting sessions. ^d^ Healthcare staff: doctors, nurses, nurses’ assistants, midwives, etc. NRT, nicotine replacement therapy.

**Table 2 ijerph-19-05815-t002:** Odds ratio (OR) for predictors of continuous abstinence at six months. Multivariable analysis is adjusted for age, sex, being disadvantaged, heavy smoking, previous quit attempts, living with a smoker, format, compliance to the GSP, and smoking cessation clinic.

Data as Observed	Crude OR		Adjusted OR ^†^	
	(95% CI)		(95% CI)	
			N = 15,197	
**Population** (N = 16,969)				
Control	1		1	
RA	1.18	(0.86–1.61)	1.28	(0.90–1.80)
OA+	0.95	(0.86–1.05)	0.92	(0.82–1.03)
**Participants**				
**Age (years)** (N = 16,969)				
18–49	1		1	
50–64	1.12	(1.04–1.19) *	1.13	(1.04–1.22) *
65+	1.07	(0.97–1.17)	0.97	(0.87–1.08)
**Sex** (N = 16,969)				
Men	1		1	
Women	0.92	(0.86–0.98) *	0.88	(0.82–0.94) *
**Disadvantaged smoker ^a^** (N = 16,352)				
No	1		1	
Yes	0.77	(0.72–0.82) *	0.79	(0.74–0.85) *
**Heavy smoker ^b^** (N = 16,613)				
No	1		1	
Yes	0.78	(0.72–0.83) *	0.74	(0.68–0.81) *
**Compliance with programme ^c^** (N = 16,747)				
No	1		1	
Yes	3.26	(3.02–3.51) *	3.29	(3.03–3.56) *
**Living with a smoker** (N = 16,835)				
No	1		1	
Yes	0.89	(0.83–0.95) *	0.91	(0.84–0.98) *
**Previous quit attempts** (N = 16,638)				
No	1			
Yes	1.08	(1.01–1.15) *		
**Free NRT or other supportive medicine** (N = 14,844)				
No	1			
Yes	0.96	(0.90–1.02)		
**Recommendation by healthcare staff ^d^** (N = 16,196)				
No	1		1	
Yes	0.90	(0.85–0.96) *	0.92	(0.85–0.99) *
**Smoking cessation clinic**				
**Setting** (N = 16,969)				
Municipality	1			
Hospital (incl. midwives)	1.08	(0.99–1.17)		
Pharmacy	0.98	(0.87–1.11)		
Other	0.92	(0.74–1.14)		
**Smoking cessation intervention**				
**Programme format** (N = 16,969)				
Group	1		1	
Individual	1.32	(1.22–1.44) *	1.23	(1.11–1.37) *
**Cluster**				
**Smoking cessation clinic**			0.05	(0.03–0.08)

RA, seropositive rheumatoid arthritis; OA, osteoarthritis; Control, non-arthritis patient. ^a^ Disadvantaged: ≤12 years of school and/or unemployed. ^b^ Heavy smoker: ≥20 pack years, Fagerström score of ≥7 points, and/or daily consumption of ≥20 cigarettes. ^c^ Compliance: attended ≥75% of the planned meeting sessions. ^d^ Healthcare staff: doctors, nurses, nurses’ assistants, midwives, etc. * Considered statistically significant if a two-sided *p*-value <0.05. ^†^ Adjusted for age, sex, being disadvantaged, heavy smoking, previous quit attempts, living with a smoker, format, compliance to the GSP, and smoking cessation clinic.

## Data Availability

Technical appendix, statistical code, and anonymised datasets will be available from the corresponding author upon reasonable request.

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
