# Peer review of "Smoking Cessation Rates among Patients with Rheumatoid Arthritis and Osteoarthritis Following the ‘Gold Standard Programme’ (GSP): A Prospective Analysis from the Danish Smoking Cessation Database"

_ijerph, 2022, doi:10.3390/ijerph19105815_

Round 1

Reviewer 1 Report

General comment: My interpretation is that this was retrospective (data was collected previously and authors reviewed charts after to create this data and manuscript). If that is the case, "prospective" should be revised to "retrospective". If it is prospective, modifications need to be made to make that clear.

Title: Change "Rheumatic" to Rheumatoid arthritis" and "musculoskeletal" to "osteoarthritis"

Abstract: change "heavy and disadvantaged smokers" to "heavy smokers, disadvantaged smokers, ..."

Introduction: remove "potentially"; sentence for reference 14 needs to be reworded to clarify

Methods: all seem appropriate, but see previous comment related to "retrospective" vs "prospective"

2.4 Intervention: second to last sentence should be "follow-up"; clarify if this is 6 months past quit date or 6 months after finishing the program

2.5 Outcomes: end of sentence should be "6 months follow-up"; Aims from Introduction should be moved to here

2.6 Data

Baseline sociodemographic data: define "short" education in this section

Follow-up Data: "personal" should be "personnel" 

2.8 Statistical Methods

3rd paragraph: move first sentence to discussion; do not need to explain the meaning and significance of MCAR and MNAR, but need to explain how you determined which category the missing data fell in

Table 1

Differences among groups are not noted in the table or results

3.3 Sensitivity Analysis: Second sentence should be reworded

Strengths and Limitations: larger materials does not make sense

Reviewer 2 Report

Aside from demographics - no detailed clinical information that can provide insight on clinical status. Multiple potential confounders.

This topic suggest a wider scope of investigation - considering the paucity of data, more rheumatic diseases should be used as comparator or a more detailed analysis should be conducted to introduce novelty and translational context. I understand that the robust sample size implies conclusions that can be extrapolated to the disease population as a whole, but this is still a very tentative conclusion.

Reviewer 3 Report

It is an interesting study,

anyway I suggest to include additional information.

Please provide a definition of smokers and ex-smokers in inclusion criteria.

Plase include the primary and secondary endpoint of the study.

Please explain how long the motivational interview and counselling were and what kind of replacement therapies were chosen , e.g patches or capsules or inhalers.

How was smoking cessation assessed , by self-reporting or exhaled CO determination? Was the five A approach applied ? it means ask-advise-assess-assist-arrange.

Please include the criteria that were used for osteoarthritis diagnosis. Why don't the authors used varenicline or citisine treatment?

What were the determinants of a reduced compliance to the program?

Among the variables a predictor of successful quitting could be symptoms worsening of osteoarthritis, could you inlcude those in the analysis?

I think that barriers and facilitators of smoking cessation should be highlighted. For example, could comorbidities influence the outcome?

I suggest to include the following references useful for both introduction and discussion regarding the use of exhaled CO as parameter for smoking cessation assessment in different settings, the variability in nicotine metabolism associated with comorbidities and the importance of smoking cessation treatment adherence.

-Biochim Biophys Acta Mol Basis Dis. 2021 Jan 1;1867(1):165990.

-Nicotine Tob Res 2018;20(10):1163-72
